# Significance of Residual Learning and Boundary Weighted Loss in Ischaemic Stroke Lesion Segmentation

**Ronnie Rajan**[*1]                                    DR.RONNIERAJAN@IITKGP.AC.IN
[1] *School of Medical Science and Technology, Indian Institute of Technology Kharagpur*

**Rachana Sathish**[*2]                                RACHANA.SATHISH@IITKGP.AC.IN
**Debdoot Sheet**[2]                                    DEBDOOT@EE.IITKGP.AC.IN
[2] *Department of Electrical Engineering, Indian Institute of Technology Kharagpur*

## Abstract

Radiologists use various imaging modalities to aid in different tasks like diagnosis of disease, lesion visualization, surgical planning and prognostic evaluation. Most of these tasks rely on the the accurate delineation of the anatomical morphology of the organ, lesion or tumor. Deep learning frameworks can be designed to facilitate automated delineation of the region of interest in such cases with high accuracy. Performance of such automated frameworks for medical image segmentation can be improved with efficient integration of information from multiple modalities aided by suitable learning strategies. In this direction, we show the effectiveness of residual network trained adversarially in addition to a boundary weighted loss. The proposed methodology is experimentally verified on the SPES-ISLES 2015 dataset for ischaemic stroke segmentation with an average Dice coefficient of 0.881 for penumbra and 0.877 for core. It was observed that addition of residual connections and boundary weighted loss improved the performance significantly.

**Keywords:** Ischaemic stroke, residual learning, adversarial training, boundary loss.

## 1. Introduction

Medical imaging tools play a vital role in aiding the physician in various tasks like estimating the anatomical morphology of organ for surgical planning, lesion visualization to evaluate damage and assessment of tumor size and spread for excision or prognostic ranking, etc. Thus precise demarcation of a lesion or organ plays a crucial role in planning and deciding life-saving therapy. One such example is the evaluation of acute ischaemic stroke and delineation of the extent of necrotic *core* in the centre of the lesion and salvageable *penumbra* around it(Dirnagl et al., 1999). Decision on thrombolytic therapy that can reverse the damage in the penumbra and thus alleviate associated symptoms, depends on the accurate estimation of extent of these lesions(ATLANTIS et al., 2004). To this regard, it is essential that all automated frameworks for automated semantic segmentation in medical imaging must include a boundary weighted loss that penalizes the misclassification of the boundary pixels heavily.

---

[*] Contributed equally

© 2019 R. Rajan, R. Sathish & D. Sheet.

## 2. Methodology and Experiments

The proposed method uses SUMNet (Nandamuri et al., 2019) as the base model for segmentation with modifications. We do not use ImageNet pre-trained weights, instead train the model from scratch with the addition of batch-normalization on the ISLES dataset[1] using three MRI sequences available in the SPES subset of the dataset, *viz.* TMax, TTP and DWI. These three sequences are concatenated into a tensor and given as the input to the network. We adopt an adversarial training strategy similar to our work accepted for publication at EMBC 2019 where we employed three relativistic discriminators operating on the segmented core, segmented penumbra and the pair respectively. In this paper we present the improvement in performance caused by the addition of residual connections (He et al., 2016) in the network along with training using a boundary weighted loss. In the modified architecture, we add residual connections in the VGG11 (Simonyan and Zisserman, 2014) like encoder after each convolutional block. The network is trained using multiple losses which include Cross-Entropy (CE) loss, Adversarial loss (Adv. loss) from the discriminators, Lovasz-Softmax (LS) loss (Berman et al., 2018) and weighted Negative Log-likelihood based boundary loss (BD) for the boundary pixels. The boundary weights for the regions of interest are computed by extracting a 2-pixel thick boundary from the ground truth annotation. This is done by performing morphological dilation followed by erosion using a structuring element of size $5 \times 5$. The pixels in the boundary are then weighed by a factor of 10.

Performance of the proposed method is compared with the following baselines. **BL1:** SUMNet trained using only CE loss, **BL2:** SUMNet with CE, LS and BD losses, **BL3:** SUMNet trained adversarially along with CE loss, **BL4:** SUMNet trained adversarially along with CE, LS and BD losses, **BL5:** Residual-SUMNet trained using only CE loss, **BL6:** Residual-SUMNet trained with CE, LS and BD losses, **BL7:** Residual-SUMNet trained adversarially along with CE loss, **Proposed:** Residual-SUMNet trained adversarially along with CE, LS and BD losses.

## 3. Results and Conclusion

Various baselines are evaluated by means of three-fold cross validation and the performance as measured by average Dice-coefficient across the folds is presented in Tab.1. And the qualitative results are shown in Fig. 1.

Table 1: Performance Evaluation of Proposed Method in terms of Dice-coefficient

|          | BL1   | BL2   | BL3   | BL4   | BL5   | BL6   | BL7   | **Proposed** |
|----------|-------|-------|-------|-------|-------|-------|-------|--------------|
| Penumbra | 0.835 | 0.838 | 0.803 | 0.841 | 0.845 | 0.844 | 0.852 | **0.881**    |
| Core     | 0.792 | 0.802 | 0.730 | 0.813 | 0.867 | 0.874 | 0.865 | **0.877**    |

It can be seen from Tab. 1, the performance increases significantly with the addition of residual connection and the boundary loss. This is also evident from the qualitative results. The residual connections in the network helps in propagation of multi-sequence information

---

1. http://www.isles-challenge.org/ISLES2015/

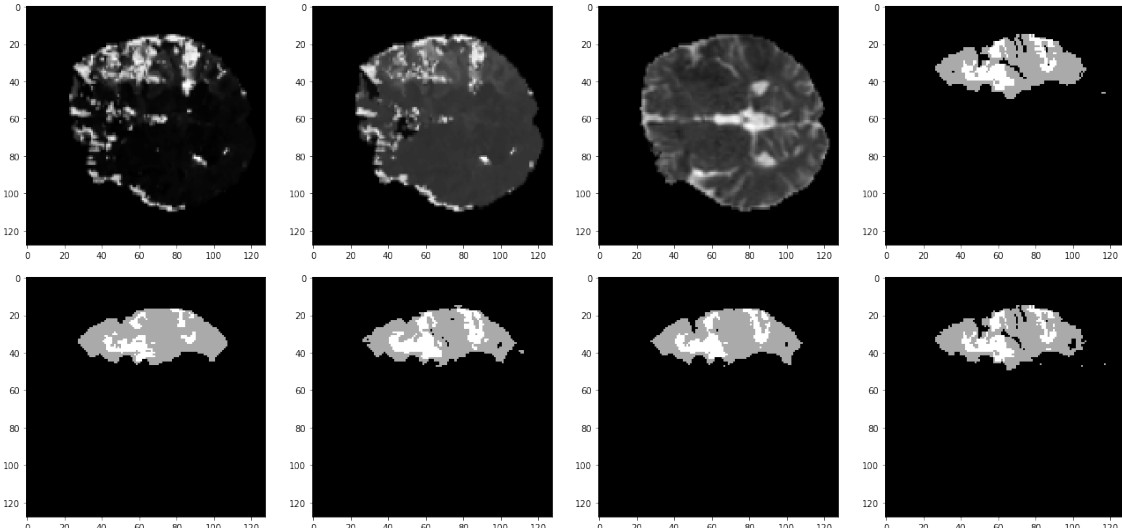

Figure 1: Sample inputs (a) Tmax, (b) TTP, (c) DWI and (d) Ground truth annotation with the segmented results for (e) BL3, (f) BL4, (g) BL7 and (h) Proposed method shown in order from top left to bottom right.

through the network. Also, the weighted boundary loss improves the segmentation along the boundary of the lesion.

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
