# OpenReview forum: "Significance of Residual Learning and Boundary Weighted Loss in Ischaemic Stroke Lesion Segmentation"
_MIDL.io/2019/Conference/Abstract — MIDL Abstract 2019_

### Official Review · AnonReviewer2 · 2019-04-27
**Not quite sure of the novelty**

**Rating:** 2
**Confidence:** 3

**Review:**

The paper emphasises on demonstrating the performance of existing techniques, including residual learning (Kaiming He, ResNet paper) and the boundary weighted loss (Olaf Ronneberger, U-net paper). The methodological contribution may be limited.

---

### Official Review · AnonReviewer1 · 2019-05-01
**Incremental improvements with residual connections and new loss function**

**Rating:** 3
**Confidence:** 2

**Review:**

This paper talks about effectiveness of residual network (trained adversarially) and a boundary weighted loss for an improved segmentation. Based on SUMNet base architecture and suggested changes on the base model, authors used SPES-ISLES 2015 dataset (ischaemic stroke segmentation) to test their proposed algorithm with an average Dice coefficient of 0.881 for penumbra and 0.877 for core.

-- the presented adversarial training strategy is from authors' EMBC 2019 work,
-- authors added residual connections (like encoder) after each convolutional block, and network was trained with multiple losses. Although there is definitely a value, incorporating dense, residual, and skip connections have been largely studied both in vision and biomedical fields.
-- It is a good attempt to have multiple loss function to improve the segmentation, it would be nice to see effect of each loss function in a more appropriate ablation study.

Overall, the paper has incremental novelties in the sense of experimental results achieving higher results than before, and showing that there is still open room to improve segmentation. Architecture design is one way, loss function is (perhaps more important) another way. Authors are taking steps towards both ways.

---

### Decision · Program_Chairs · 2019-05-06
**Acceptance Decision**

Accept